**Data Availability Statement:** All relevant data are within the manuscript and its Supporting information files.

**Funding:** The author(s) received no specific funding for this work.

# The simplified hybrid model based on BP to predict the reference crop evapotranspiration in Southwest China

**Zhenhua Zhao**[iD]**[1]\*, Guohua Feng[2], Jing Zhang[3]**

**1** College of International Education, Bohai University, Jinzhou, Liaoning Province, China, **2** Foundation Department, Liaoning Agriculture and Economy School, Jinzhou, Liaoning Province, China, **3** College of Mechanical and Electronic Engineering, Wuhan University of Technology, Wuhan, Hubei Province, China

\* lawrence@bhu.edu.cn

## Abstract

The accurate prediction of reference crop evapotranspiration is of great significance to climate research and regional agricultural water management. In order to realize the high-precision prediction of $ET_O$ in the absence of meteorological data, this study use XGBoost to select key influencing factors and BP algorithm to construct $ET_O$ prediction model of 12 meteorological stations in South West China in this study. ACO, CSO and CS algorithms are used to optimize the model and improve the adaptability of the model. The results show that $T_{max}$, n and Ra can be used as the input combination of $ET_O$ model construction, and $T_{max}$ is the primary factor affecting $ET_O$. $ET_O$ model constructed by BP algorithm has good goodness of fit with the $ET_O$ calculated by FAO-56 PM and ACO, CSO and CS have significant optimization effect on BP algorithm, among which CSO algorithm has the best optimization ability on BP, with RMSE, $R^2$, MAE, NSE, GPI ranging 0.200–0.377, 0.932–0.984, 0.140–0.261, 0.920–0.984, 1.472–2.000, GPI ranking is 1–23. Therefore, the input combination ($T_{max}$, n and Ra) and CSO-BP model are recommended as a simplified model for $ET_O$ prediction in Southwest China.

## Introduction

Reference crop evapotranspiration ($ET_O$) is an important parameter of ecological water cycle and a key value to calculate actual evapotranspiration. Under the influence of the intensification of Walker circulation caused by La Nina, the climate change in Southeast Asia is abnormal, which is likely to cause floods and droughts, and the agricultural water balance will also be potentially affected. Therefore, accurate and efficient prediction of $ET_O$ is of great significance to farmland ecological management and irrigation decision-making.

Many models have been developed to predict $ET_O$, such as Hargreaves [1], Turc [2], FAO-56 Penman-Monteith (FAO-56 PM) [3], etc. FAO-56 PM considers all relevant factors (radiation, average wind speed, humidity, maximum or minimum temperature, etc.) and is recognized as the standard formula for $ET_O$ calculation. However, due to the uneven distribution

**Competing interests:** The authors have declared that no competing interests exist.

and inconsistent scale of meteorological stations, it is difficult to obtain all meteorological parameters used to calculate $ET_O$, which makes it difficult to accurately calculate $ET_O$.

Many scholars have proposed some empirical models with single input [4–7]. Tabari et al. [8] (2011) conducted a study in Iran to compare the adaptability of 31 empirical models under humid conditions. By comparing the model accuracy, it is found that the model based on temperature and radiation (especially HS) has the closest estimation value to the standard PM formula. Djaman et al. [9] evaluated the $ET_O$ simulation capability of multiple empirical formulas by using the meteorological data of Senegal River Valley. The results show that mass transfer models have the best performance. The accuracy of empirical models is generally low, and different climatic conditions will have a great impact on the results of the model.

In recent years, machine learning has become a hot topic in the field of data analysis. It is widely used to deal with nonlinear and complex problems and has great advantages in predicting $ET_O$ [10, 11], such as artificial neural network (ANN), extreme learning machine (ELM), support vector machine (SVM), etc. [12, 13]. Min et al. [14] proposed several machine learning models (SVM-HS and SVM ROM, etc.) to simulate empirical formulas. When the input factors are the same, SVM model performs better than empirical model. Extreme Learning Machine (ELM) is also an efficient machine learning algorithm [15]. Its performance in estimating $ET_O$ is better than Hargreaves and ANN model.

Among various intelligent algorithms, Error Back Propagation (BP) algorithm has good generalization ability and adaptive ability [16, 17]. However, when machine learning algorithm constructs a model, model parameters are difficult to reach the optimality. Optimization algorithm has a wide range of applications in the process of parameter optimization, and many scholars use optimization algorithms to improve the accuracy of models in recent years. Fang et al. [18] used the fruit fly optimization algorithm (FOA) to optimize the $ET_O$ constructed by generalized regression neural networks (GRNN) algorithm, and obtained a more efficient and adaptive prediction model. Dong [19] compared Coupling Bat algorithm-categorical features support (Bat-CB) hybrid algorithm and CB's adaptability to $ET_O$ prediction. The results show that Bat-CB has better robust stability.

There are many difficulties in the actual collection of meteorological data, which makes it difficult to obtain all meteorological parameters used to calculate $ET_O$, therefore many studies select input parameters of models based on previous research experience, lacking clear theoretical basis. Some scholars used linear analysis methods, for example, determine the dominant factors of $ET_O$ model based on path analysis theory, principal component analysis and factor analysis [20, 21]. Compared with linear analysis methods, machine learning algorithms can better analyze nonlinear problems. Therefore, this study uses machine learning algorithm to screen the factors with high contribution to the prediction results of the model.

The main objectives of this study are: (1) XGBoost algorithm was used to analyze the contribution rate of meteorological factors to $ET_O$, and the combination of few factors with a large impact on $ET_O$ was obtained. (2) BP algorithm and optimized hybrid algorithm (ACO-BP, CSO-BP, CS-BP) were used to build the $ET_O$ prediction model. (3) The accuracy and adaptability of $ET_O$ model were evaluated in southwest China.

## Materials and methods

### Data sources

Southwest China is located in 91°21′-112°04′E and 20°54′-34°19′N, which consists of Chongqing, Sichuan, Guizhou and Yunnan. The Tropic of Cancer runs through southern Yunnan. It has various terrain types, including plateau, mountain, hill, basin and plain, with Karst and volcanic landforms as well. It is at the junction of the first and second steps in our country and

the terrain is higher in the northwest and lower in the southeast. The elevation of Qingzang Plateau is above 4000m, Yunnan-Guizhou Plateau is below 2000m, but Sichuan Basin is below 500m. The undulating terrain and complex direction of the mountains have a significant impact on the climate. There are obvious differences in the horizontal distribution of temperature. In the same area, the vertical distribution of temperature is also prominent. Affected by the Pacific southeast monsoon and Indian southwest monsoon, it has a tropical and subtropical monsoon climate. Southwest of China is rich in water sources, but the regional distribution of precipitation is unbalanced. The rainfall on the windward slope can exceed 2000mm, however, the leeward and valley bottom are only 600-700mm. The annual precipitation of Southwest China is <900mm. Relevant research shows that Southwest of China has shown a trend of high temperature and little rain in recent years. From 1961 to 2017, the annual average temperature in Southwest China increased at a rate of 0.16°C/10 years. The rising trend of average temperature in autumn and winter is most obvious. Rainfall decreases in the east and increases in the west. The rainy season and autumn rainy period both show a shortening trend. From 1961 to 2017, the annual precipitation in the southwest region decreased at a rate of 9.4mm/10 years.

The study considered 12 stations [Liuzhou (C1), Tongren (C2), Baise (C3), Nanning (C4), Baoshan (C5), Yuxi (C6), Mengzi (C7), Barkam (C8), Yaan (C9), Bazhong (C10), Kaili (C11), Liangping (C12)] in Southwest China.

The meteorological data in this study were daily dataset during 1960–2019 comes from China Meteorological Data Network (http://data.cma.cn/), including sunshine duration (n), average air temperature (Ta), maximum air temperature ($T_{max}$), minimum air temperature ($T_{min}$), wind speed (Wind) and relative humidity (RH) extraterrestrial solar radiation (Ra).

## FAO-56 PM

The daily reference evapotranspiration was calculated by FAO-56 PM equation:

$$ET_0 = \frac{0.408\Delta(R_n - G) + \gamma \frac{900}{T_{mean}+273} U_2(e_s - e_a)}{\Delta + \gamma(1 + 0.34U_2)} \tag{1}$$

where $\Delta$ is Saturated water pressure—Slope of temperature curve (kPa/°C), $R_n$ is the net solar radiation (MJ m−2day-1) $G$ is the soil heat flux density (MJ m−2day-1), $\gamma$ is the psychrometric constant (kPa °C-1). $T_{mean}$ is the mean air temperature (°C), $U_2$ is the wind speed at 2 m (M/s), $e_s$ is saturated vapor pressure (kPa), $e_a$ is the actual vapor pressure (kPa).

## Different machine learning for predicting daily reference crop evapotranspiration

**Back-Propagation neural network (BP).** BP neural network [22] is the most traditional neural network. But compared with other traditional models, it has better persistence and timely prediction. BP neural network is to analyze the error between the training result and the expected result, so as to modify the weight and threshold, and get a model that can output the same as the expected result. BP neural network is composed of input layer, hidden layer and output layer. It is a multi-layer forward network based on error direction propagation. The neurons in each layer are fully connected, and the neurons in the same layer are not connected. By collecting and returning the errors generated by the system in the process of simulation, we can use these errors to adjust the weight of neurons and generate an artificial neural network system that can simulate the original problem (Fig 1). The parameter conditions of the algorithm are set to: training times (net.trainParam.epochs = 1000), learning rate (net.

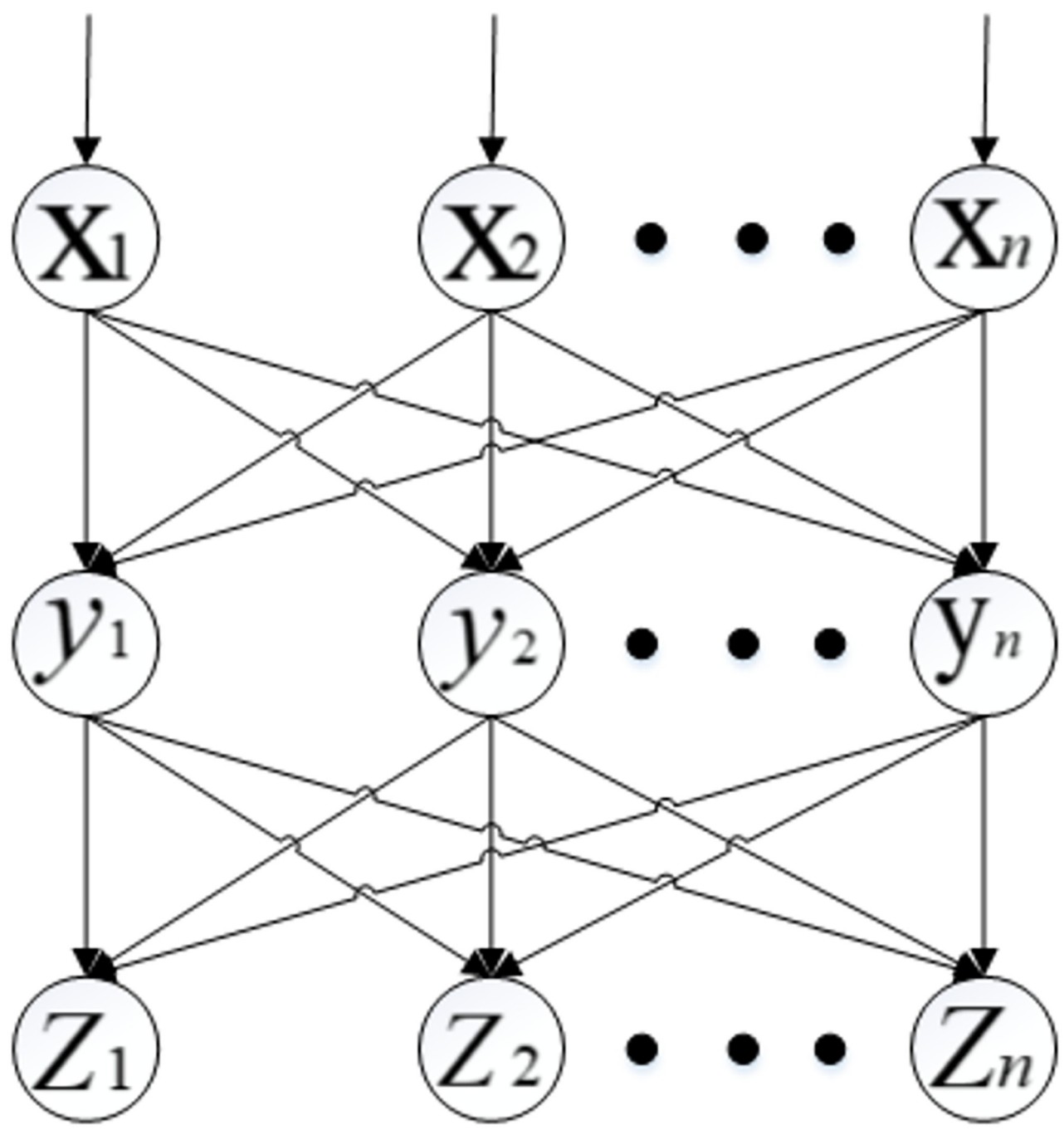

**Fig 1. BP neural network model diagram.**

trainParam.lr = 0.01), Minimum error of training goal (net.trainParam.goal = 0.00001), momentum factor (net.trainParam.mc = 0.01).

Given training set D = {($x_1$,$y_1$),($x_2$,$y_2$...($x_n$,$y_n$)} where $x_n \epsilon R^d$, $y_n \epsilon R^l$, Therefore, the weight from the input layer to the hidden layer is set to $v_i h$, the threshold of the H-th neuron in the

hidden layer is set to $y_h$, and weight from hidden layer to output layer, is set to $\omega_{hj}$

$$\Delta\omega_{hj} = -\eta\,\frac{\partial Ek}{\partial\omega_{hj}} \tag{2}$$

Input layer to hidden layer:

$$\alpha_h = \sum_{i=1}^{d} v_{ih}x_i \tag{3}$$

Activation function through hidden layer:

$$bh = f(\alpha h - \gamma h) \tag{4}$$

Hidden layer to output layer:

$$\beta_j = \sum_{h=1}^{q} \omega_{hj} * b_h \tag{5}$$

Activation function through output layer:

$$y_j^k = f(\beta_j - \theta_j) \tag{6}$$

Error:

$$Ek = \frac{1}{2}\sum_{j=1}^{l}(y_j^{k'} - y_j^k)^2 \tag{7}$$

**Ant colony optimization algorithm (ACO).** ACO algorithm [23, 24] is a part of swarm intelligence, and a global optimization algorithm. It was first proposed by Marco Dorigo et al. in 1991. It is a probabilistic algorithm used to find the optimal path in the graph. The basic idea of ant colony algorithm comes from the shortest path principle of ant foraging in nature. When looking for food, the ant colony exchanges foraging information by secreting a biological hormone called pheromone. If the path is shorter and the pheromone concentration is higher, more and more ants will choose a shorter path. Finally, they can find the shortest path from the food source to the nest without any hint, and adaptively search the new best path after the surrounding environment changes, so as to find the target quickly. The basic idea of ACO algorithm is as follows:

Suppose there are s parameters in the network. The neural network parameter Pi (1≤i≤s) is set to N random non-zero values to form a set IP. Then simulate the foraging behavior of ant colony. When ants start from the ant nest to find food, each ant starts from the first set and selects an element from each set according to a certain probability and according to the information state of each element in the set. When the ant completes the selection of elements in all sets, it reaches the food source and returns to the ant nest according to the original path. At the same time, the pheromone of the selected element in the set is updated according to the following formula (8), and the process is repeated. When all ants converge to the same path, it means that the optimal solution of network parameters is found.

$$\boldsymbol{\tau_j(I_{Pi})}(t+n) = \boldsymbol{\rho\tau_j(I_{Pi})}(t) + \boldsymbol{\Delta\tau_j(I_{Pi})} \tag{8}$$

$\boldsymbol{\tau_j}(\boldsymbol{I_{pi}})(t)$ represents the amount of information on element J in set $\boldsymbol{I_{pi}}$ **at** t time, ρ Represents the maintenance factor of pheromone, $\boldsymbol{\Delta\tau_j}(\boldsymbol{I_{pi}})$ represents the sum of pheromones

released by all ants on element J in set $\mathbf{I_{pi}}$, the expression is

$$\Delta\tau_j^k(\mathbf{I_{Pi}}) = \begin{cases} \frac{Q}{e^k} & \text{If the } k-\text{th ant selects } P_j(\mathbf{I_{Pi}}) \text{ in this cycle} \\ 0 & \text{otherwise} \end{cases} \tag{9}$$

Where, $\mathbf{e^k}$ represents a set of weights selected by the k-th ant is used as the output error of the weights of the neural network, where o and $\mathbf{o_q}$ represent the actual output and expected output of the neural network. The parameter conditions of the algorithm are set to: population size (popsize = 10), maximum Generation (maxgen = 50), pheromone polatility (rou = 0.9), transition probabilities constant (p0 = 0.2).

**Cat Swarm Optimization (CSO).** CSO optimization algorithm [25, 26] was first proposed by Shu An Chu et al. in 2006. It is a new swarm optimization algorithm based on the predation strategy of cats, and is generally used to find the optimal solution. Cat swarm algorithm includes two important simulation processes, "search mode" and "tracking mode". The cat's pattern of laziness and looking around is called search pattern; the state of a cat when tracking a dynamic target is called tracking mode. The steps of cat swarm algorithm are as follows:

a. Determine the number of individuals involved in the optimization calculation, that is, the number of cats. Each cat has a d-dimensional position coordinate value, and $x_{i,d}$ represents the position coordinate value of the $i_{th}$ cat in the $d_{th}$ dimension.

b. Randomly initialize the velocity $v_{i,d}$ for each one-dimensional position.

c. The fitness function value of each cat is evaluated, and the cat with the optimal fitness function value is regarded as the local optimal cat.

d. Cats were randomly assigned to search mode and tracking mode according to the mixture ratio.

e. Their fitness is calculated according to the fitness function value, and the best solution in the current population is retained.

f. This method is used for iterative calculation until the preset number of iterations is reached.

The parameter conditions of the algorithm are set to: population size(popsize = 10), maximum iterations(npop = 50), mixture ratio (MR = 0.3). For cats in search mode, the following necessary parameters are defined: Search Memory Pool (SMP), Seeking Range of selected Dimension (SRD), Counts of Dimension to Change (CDC) and Mixture Ratio (MR). The operation process is as follows:

1. Copy the SMP copy of the cat in the search mode.

2. For each individual copy in the memory pool, determine which dimension positions need to be changed according to the CDC value, and randomly increase or decrease the SRD ratio of the dimension position value that needs to be changed.

3. The fitness values of all candidate solutions in the memory pool are calculated respectively.

4. Select the candidate point with the highest fitness value to replace the current cat's position and complete the cat's position update.

Tracking mode simulates the tracking target behavior in cat hunting behavior, which is equivalent to local search in optimization problem. This model updates the position of the cat by changing the speed (i.e. eigenvalue) of each dimension of the cat to achieve the optimal

position. Suppose the position coordinates and speeds of the cat are expressed as:

$$x_i = (x_{i,1}, \ x_{i,2}, \ x_{i,3}, \dots, x_{i,m}) \sim i = 1, 2, 3, \dots, m \tag{10}$$

$$v_i = (v_{i,1}, \ v_{i,2}, \ v_{i,3}, \dots, v_{i,m}) \sim i = 1, 2, 3, \dots m \tag{11}$$

The cat with local optimal solution during the operation of cat swarm algorithm is expressed as:

$$x_b = (x_{b,1}, \ x_{b,2}, \ x_{b,3}, \dots, x_{b,m}) \sim i = 1, 2, 3, \dots, m \tag{12}$$

For a cat in tracking mode, its moving position is determined according to its speed. First determine the update speed:

$$v_{k,d} = v_{k,d} + r_1 \times c_1 \times (x_{best,d} - x_{k,d}) \tag{13}$$

where d = 1,2,...,M
Where $r_1$, $c_1$ is the adjustment parameter and takes a constant.
The position of a cat changes by speed:

$$x_{k,d} = x_{k,d} + v_{k,d} \tag{14}$$

**Cuckoo optimization algorithm (CS).** CS optimization algorithm [27] simulates the nesting habit of cuckoo. It relies on other birds to hatch and brood its own offspring. The CS algorithm assumes the following ideal states: Each cuckoo lays only one egg at a time, and a nest is randomly selected for storage; In the process of nest searching, the nest with the best eggs will be reserved for the next generation. The number of available nests is fixed, and let the probability of foreign eggs being found in the nest be P, P∈[0,1].

In the cuckoo search algorithm, the cuckoo's lévy flight route to find the nest and the cuckoo's random flight route to avoid its eggs being found by the nest owner are two important routes of the algorithm. Lévy flight, which is used to optimize the search, is one of the most effective target finding methods in CS algorithm. The parameter conditions of the algorithm are set to: initial population size (PopulationSize_Data = 30), probability of cuckoo eggs being found (pa = 0.25), step control amount (cs_alpha = 1.0).

The formula for Lévy flight is as follows:

$$X_i^{(t+1)} = x_i^t + \alpha \oplus Levy(\beta) \tag{15}$$

Where, $X_i^{(t+1)}$ represents the position of the bird's nest, α represents the step control vector, $Levy(\beta)$ represents Levi's random search path. As shown in the formula, the schematic diagram of Lévy flight is as follows:

$$Levy(\beta) \sim \mu = t^{-\beta} \quad 1 < \beta \le 3 \tag{16}$$

CS algorithm calculates the fitness value of the objective function, and if the location of the next generation nest is better, this location will be updated. After the location is updated, compare the probability P of being found by the host with the random number R, and If R>P, then $X_i^{(t+1)}$ changes randomly. On the contrary, it does not change. Finally, a group of nest positions with better test values shall be reserved. The technical process of this study is in Fig 2.

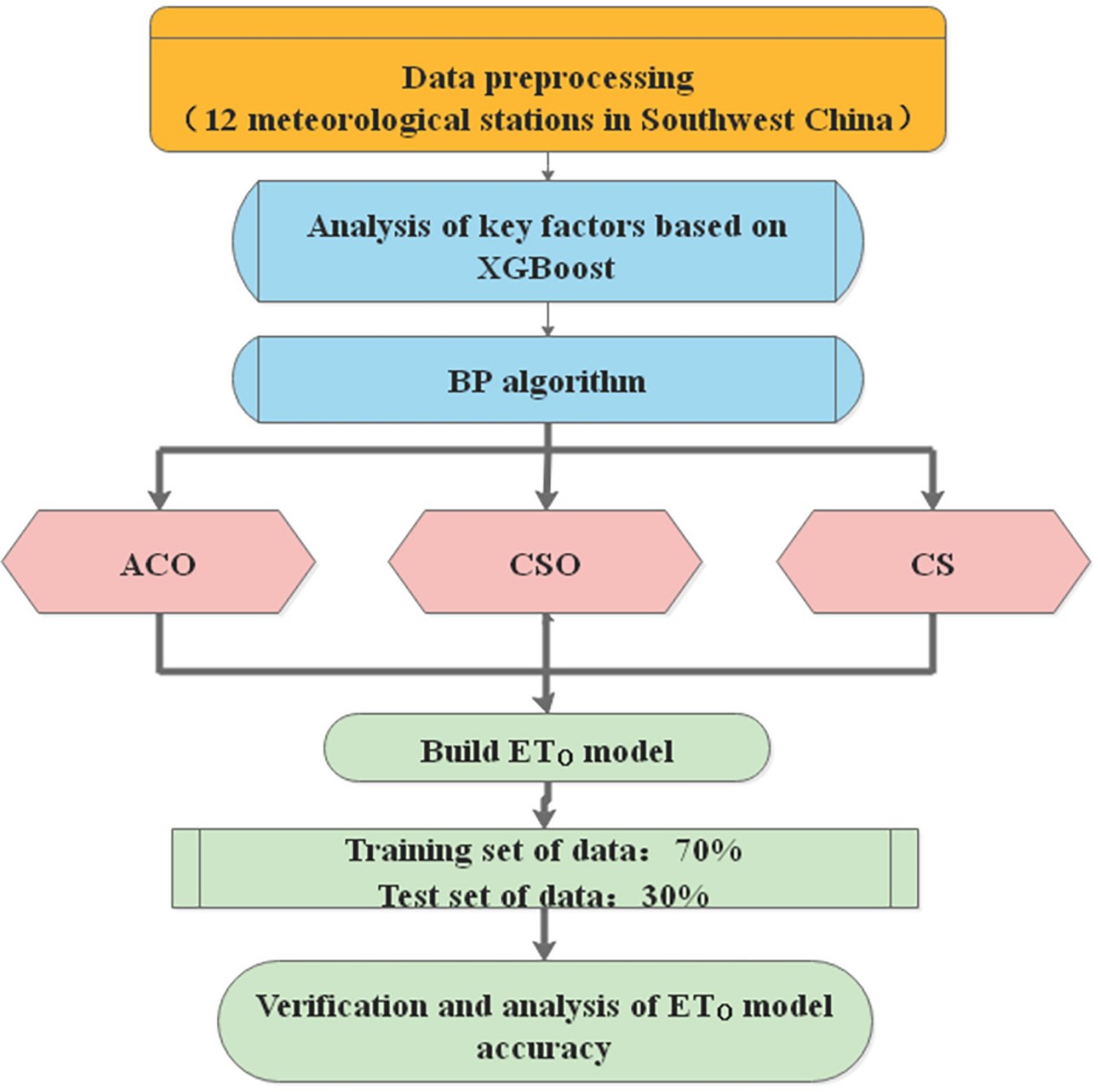

**Fig 2. The technical process of this study.**

### Model prediction evaluation

The coefficient of determination ($R^2$), relative root mean square error (RMSE), mean absolute error (MAE), and Nash-Sutcliffe coefficient (NSE) and overall evaluation index (GPI) were used to evaluate performances of the models [28].

$$R^2 = \frac{\left[\sum_{i=1}^{n} Di - \bar{D}(Ei - \bar{E})\right]^2}{\sum_{i=1}^{n} Di - \bar{D}^2 \sum_{i=1}^{n} (Ei - \bar{E})^2} \tag{17}$$

$$RMSE = \sqrt{\frac{1}{n}\sum\nolimits_{i=1}^{n}\left(Di - Ei\right)^2} \qquad (18)$$

$$NSE = 1 - \frac{\sum_{i=1}^{n}\left(Di - Ei\right)^2}{\sum_{i=1}^{n}\left(Yi - \bar{E}\right)^2} \qquad (19)$$

$$MAE = \frac{1}{n}\sum\nolimits_{i=1}^{n}\left|Di - Ei\right| \qquad (20)$$

$$GPI = \alpha_j \sum\nolimits_{i=1}^{4}\left(S_j - \bar{T}_j\right) \qquad (21)$$

where $D_i$ and $E_i$ are the simulated and measured values, respectively; n is the number of measured values; $\bar{D}$ and $\bar{E}$ are the means of the simulated and measured values, respectively. $T_j$ is the normalized value of RMSE, MAE, $R^2$, NSE, $\bar{T}_j$ is the median of the corresponding parameter, when $S_j$ is RMSE and MAE, $\alpha_j$ is -1, otherwise take 1.

## Results

### Analysis of key factors based on XGBoost algorithm

The meteorological parameters used to build the $ET_O$ prediction model were strongly coupled. It is difficult to directly determine the impact of a single factor on model construction. In this study, XGBoost algorithm is applied to determine the essential factors. Fig 3 shows the influence degree of seven meteorological factors on $ET_O$ prediction.

As can be seen from the figure, $T_{max}$ is the primary factor affecting $ET_O$ calculation. The importance range of 11 stations is 0.405–0.704, which is much higher than other meteorological factors, indicating that temperature is significantly correlated with $ET_O$. The factor ranking second in importance is n, and the range of importance is 0.223–0.297. Ra is one of the important factors in calculating evapotranspiration. In most sites, the importance of Ra is second only to n, and the importance result is 0.038–0.167. However, in Yuxi, RH has a more significant effect than Ra. The air humidity in Yuxi varies strongly in different seasons. It rains frequently from May to October, and heavy rainstorms are mostly concentrated from June to August. The "single point rainstorm" with small range and high intensity occurs frequently, which may affect the correlation calculation between humidity and $ET_O$ by the algorithm. The sum of the importance of the top three factors obtained by the algorithm ranges from 0.829 to 0.982. In the FAO-56 calculation formula, $T_{max}$, n and Ra are also important constituent parameters, which shows strong rationality.

In general, the impact of meteorological factors on $ET_O$ is consistent at all stations. The sum of the importance of $T_{max}$, n and Ra accounts for more than 82.9% of all factors, which can be used as a representative factor to construct the input combination. Therefore, this study takes these three factors as the input combination of subsequent $ET_O$ models.

### Statistical performance of $ET_O$ models

The input combination ($T_{max}$, n and Ra) obtained through key factor analysis is used to construct $ET_O$ prediction model. In this study, BP algorithm and three optimization algorithms are used to build $ET_O$ prediction model. Fig 4 shows the accuracy of each model, and the specific accuracy index of each model is shown in Table 1. It can be seen from the table that the

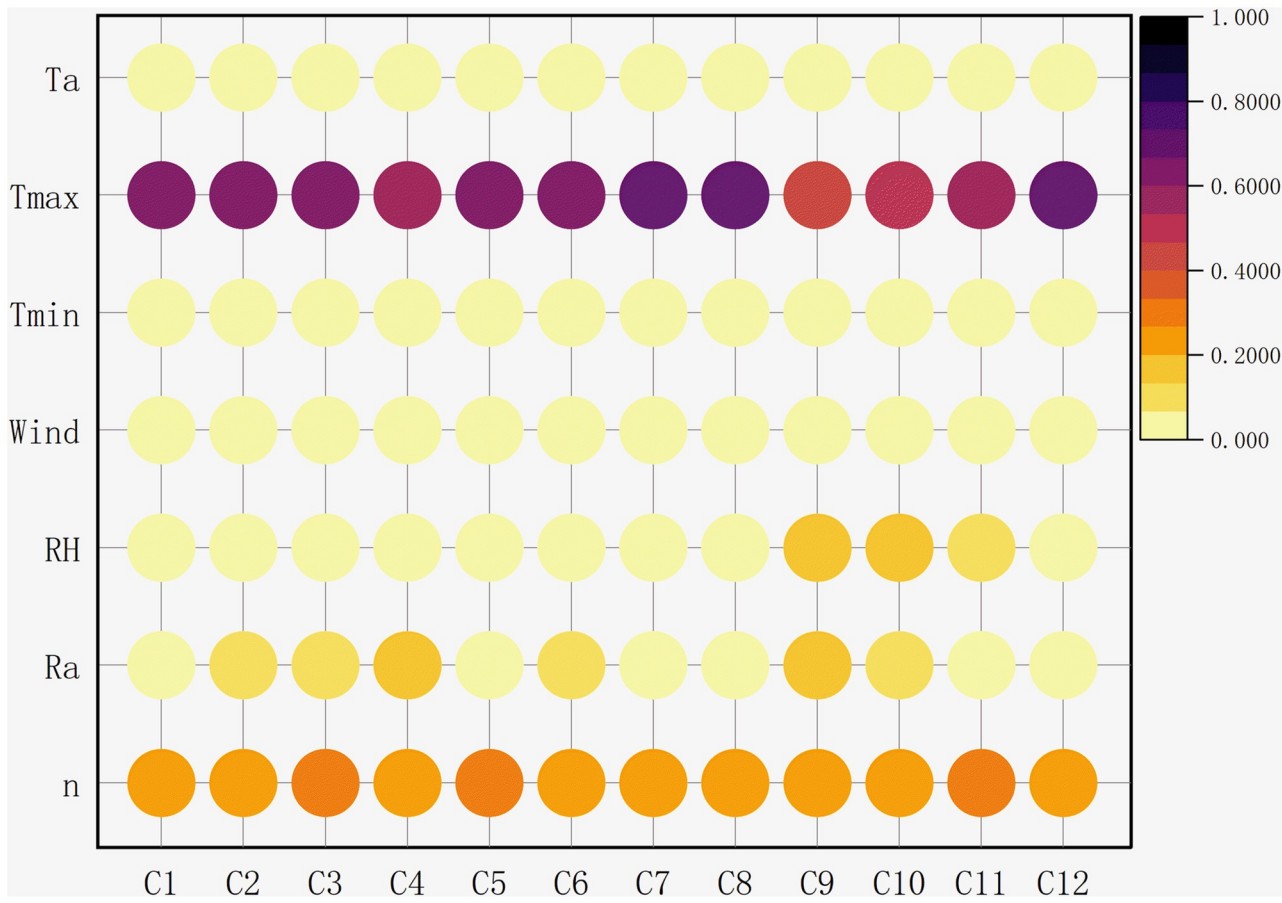

**Fig 3. Importance of meteorological factors to ET$_O$ at different stations.**

accuracy of ET$_O$ model constructed by BP algorithm is satisfactory, with RMSE, R$^2$, MAE, NSE ranging 0.488–0.779, 0.499–0.930, 0.386–0.759, 0.400–0.905. The three optimization algorithms have good optimization effect on BP model. Among them, CSO-BP has the highest accuracy, with RMSE, R$^2$, MAE, NSE ranging 0.200–0.0377, 0.932–0.984, 0.140–0.261, 0.920–0.984. The accuracy of CS-BP and ACO-BP is greatly improved compared with the unoptimized BP model, with RMSE, R$^2$, MAE, NSE ranging 0.209–0.387, 0.930–0.982, 0.149–0.265, 0.915–0.982 (CS-BP), 0.422–1.131, 0.682–0.962, 0.287–0.945, 0.352–0.899 (ACO-BP).

The comprehensive evaluation index GPI performance of ET$_O$ model is shown in Table 2. It can still be seen that CSO-BP has the best fitting ability. The GPI range is 1.472–2.000 and the GPI ranking range is 1–23. The performance of the four models in GPI is the same as that of the previous four evaluation indexes.

The performance of ET$_O$ model in each station is shown in Fig 5. In Bazhong station, the accuracy of BP algorithm is the highest, GPI is 1.108 and GPI ranking is 26. In Baoshan, Yuxi and Barkam stations, the accuracy of BP model is low. ACO-BP is superior to BP model in most stations, but it has poor adaptability in Ya'an and Bazhong stations. The performance of CSO-BP and CS-BP algorithms in 12 stations is very similar. It can be seen that the adaptability of the two models in different sites is very satisfactory. They perform best in Liangping site (GPI = 2.000, 1.951; ranking 1 and 2) and slightly worse in Yuxi site (GPI = 1.472, 1.436; ranking 23 and 24). Among all ET$_O$ models, CSO-BP model has the best accuracy advantage in

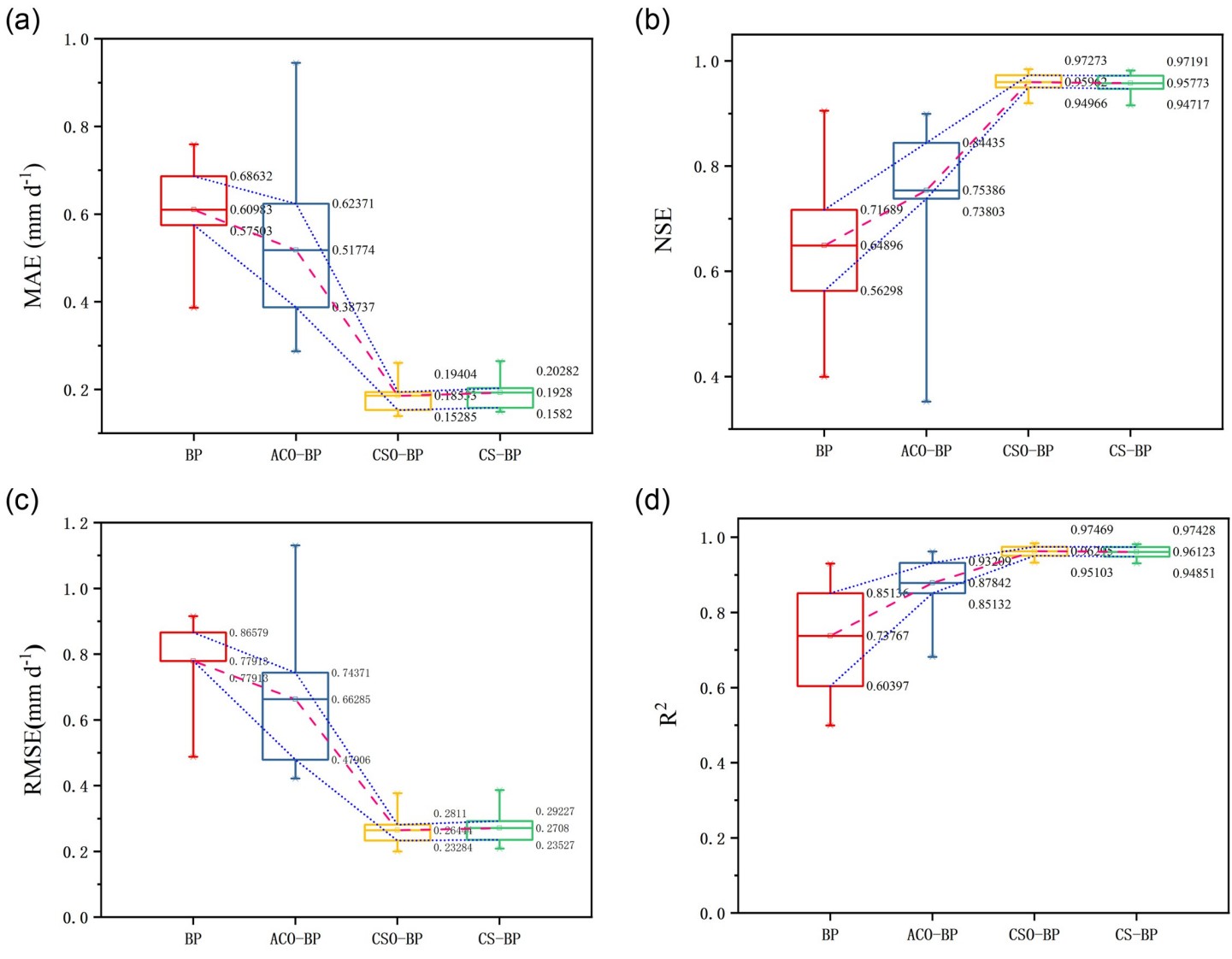

**Fig 4. (a-d) Comparison of different $ET_O$ models.**

estimating $ET_O$ at each site. The GPI range is 1.472–2.000 and the ranking range is 1–23. And using the combination of only three factors as the input factor to build the model, CSO-BP can maintain good stability and high accuracy.

## Discussion

Using comprehensive meteorological factors to calculate $ET_O$ can obtain the highest accuracy, but restricted by the actual situation of meteorological stations, meteorological data are sometimes missing, and the determination of some parameters takes time and manpower. Therefore, it is necessary to explore an input combination with only a few factors. Many scholars have explored the model of simplifying input factors, through trial-and-error method assisted by computer software [29, 30]. However, these methods have too much workload and do not have universality.

**Table 1. Accuracy indicators of the four ET$_O$ models.**

| Site | BP | | | | ACO-BP | | | | CSO-BP | | | | CS-BP | | | |
|---|---|---|---|---|---|---|---|---|---|---|---|---|---|---|---|---|
| | RMSE | $R^2$ | MAE | NSE | RMSE | $R^2$ | MAE | NSE | RMSE | $R^2$ | MAE | NSE | RMSE | $R^2$ | MAE | NSE |
| Liuzhou | 0.844 | 0.930 | 0.628 | 0.663 | 0.537 | 0.962 | 0.397 | 0.863 | 0.252 | 0.972 | 0.186 | 0.970 | 0.258 | 0.971 | 0.190 | 0.969 |
| Tongren | 0.816 | 0.759 | 0.652 | 0.658 | 0.722 | 0.954 | 0.484 | 0.732 | 0.230 | 0.975 | 0.152 | 0.973 | 0.234 | 0.974 | 0.158 | 0.972 |
| Baise | 0.590 | 0.851 | 0.482 | 0.833 | 0.601 | 0.829 | 0.484 | 0.827 | 0.282 | 0.964 | 0.203 | 0.962 | 0.292 | 0.961 | 0.212 | 0.959 |
| Nanning | 0.845 | 0.604 | 0.670 | 0.600 | 0.527 | 0.864 | 0.443 | 0.844 | 0.205 | 0.977 | 0.153 | 0.976 | 0.209 | 0.976 | 0.155 | 0.976 |
| Baoshan | 0.869 | 0.500 | 0.686 | 0.405 | 0.479 | 0.933 | 0.354 | 0.819 | 0.281 | 0.950 | 0.189 | 0.938 | 0.292 | 0.947 | 0.203 | 0.933 |
| Yuxi | 0.916 | 0.582 | 0.724 | 0.525 | 0.422 | 0.932 | 0.287 | 0.899 | 0.377 | 0.935 | 0.248 | 0.920 | 0.387 | 0.930 | 0.255 | 0.915 |
| Mengzi | 0.866 | 0.643 | 0.693 | 0.563 | 0.473 | 0.923 | 0.387 | 0.870 | 0.348 | 0.932 | 0.261 | 0.929 | 0.347 | 0.932 | 0.265 | 0.930 |
| Barkam | 0.895 | 0.546 | 0.759 | 0.400 | 0.467 | 0.887 | 0.374 | 0.837 | 0.259 | 0.951 | 0.186 | 0.950 | 0.266 | 0.949 | 0.194 | 0.947 |
| Yaan | 0.589 | 0.841 | 0.447 | 0.813 | 1.097 | 0.682 | 0.945 | 0.352 | 0.233 | 0.971 | 0.170 | 0.971 | 0.240 | 0.969 | 0.179 | 0.969 |
| Bazhong | 0.488 | 0.911 | 0.386 | 0.905 | 1.131 | 0.804 | 0.776 | 0.491 | 0.233 | 0.979 | 0.145 | 0.978 | 0.235 | 0.979 | 0.149 | 0.978 |
| Kaili | 0.788 | 0.898 | 0.575 | 0.706 | 0.744 | 0.920 | 0.659 | 0.738 | 0.273 | 0.965 | 0.194 | 0.965 | 0.275 | 0.965 | 0.196 | 0.964 |
| Liangping | 0.845 | 0.788 | 0.616 | 0.717 | 0.755 | 0.851 | 0.624 | 0.774 | 0.200 | 0.984 | 0.140 | 0.984 | 0.215 | 0.982 | 0.158 | 0.982 |

The algorithm quantifies the characteristic correlation between meteorological factors and ET$_O$, which is more efficient. Xing et al. [20] used path to analyze the direct impact of various meteorological factors on the path coefficient and factors of ET$_O$, and determined the leading factors for constructing the prediction model. Machine learning algorithms have better solutions to nonlinear problems than mathematical statistics. Some scholars use the random forest algorithm to recognize the optimal factor combination, which has a better conclusion than the linear algorithm [31].

This paper calculates the characteristic importance of seven meteorological factors on ET$_O$ model construction through XGBoost algorithm, and uses a few factors that have a great impact on ET$_O$ as the input combination of model construction. In the analysis of characteristic importance of meteorological factors, T$_{max}$ is considered to be the primary factor affecting ET$_O$, with the highest importance among all factors (0.405–0.704). In the construction of evapotranspiration model, Wu et al. found that the factor combination based on T$_{max}$ was input into the model, and the prediction accuracy was the highest [32]. This is similar to the conclusion of this paper.

**Table 2. GPI value and GPI ranking of each site.**

| Site | BP | | ACO-BP | | CSO-BP | | CS-BP | |
|---|---|---|---|---|---|---|---|---|
| | GPI | Ranking | GPI | Ranking | GPI | Ranking | GPI | Ranking |
| Liuzhou | 0.082 | 39 | 1.081 | 28 | 1.839 | 11 | 1.824 | 12 |
| Tongren | -0.279 | 41 | 0.550 | 35 | 1.914 | 7 | 1.900 | 8 |
| Baise | 0.642 | 32 | 0.573 | 34 | 1.757 | 15 | 1.723 | 18 |
| Nanning | -0.744 | 42 | 0.803 | 31 | 1.950 | 3 | 1.940 | 4 |
| Baoshan | -1.314 | 46 | 1.066 | 29 | 1.706 | 19 | 1.663 | 20 |
| Yuxi | -1.050 | 45 | 1.336 | 25 | 1.472 | 23 | 1.436 | 24 |
| Mengzi | -0.774 | 43 | 1.092 | 27 | 1.496 | 21 | 1.493 | 22 |
| Barkam | -1.345 | 47 | 0.989 | 30 | 1.755 | 16 | 1.730 | 17 |
| Yaan | 0.633 | 33 | -1.588 | 48 | 1.877 | 9 | 1.851 | 10 |
| Bazhong | 1.108 | 26 | -0.943 | 44 | 1.938 | 5 | 1.929 | 6 |
| Kaili | 0.209 | 37 | 0.249 | 36 | 1.782 | 13 | 1.777 | 14 |
| Liangping | -0.112 | 40 | 0.196 | 38 | 2.000 | 1 | 1.951 | 2 |

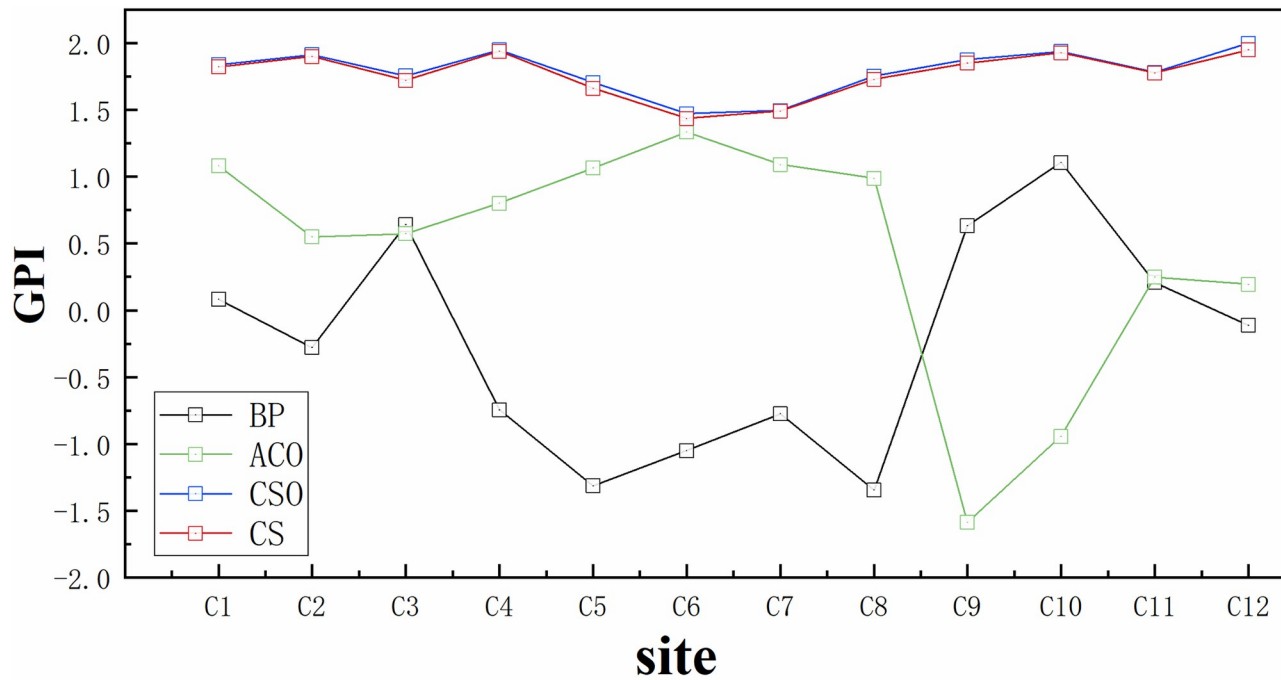

**Fig 5. GPI performance of the four models at each site.**

In addition to $T_{max}$, the main factors affecting the $ET_O$ model include n and Ra. The sum of the importance of the three factors is 0.829–0.982, which is the main factor affecting the model construction.

This paper uses BP and hybrid algorithm (ACO-BP, CSO-BP, CS-BP) to build $ET_O$ model. The input of the model adopts the combination of $T_{max}$, n and Ra. The results show that the $ET_O$ model based on machine learning algorithm is acceptable. Based on the optimization algorithm, the prediction results of the optimized hybrid model have better coupling with the standard value. ACO optimization algorithm has strong robustness in optimization performance, but ACO-BP algorithm has the lowest accuracy among all hybrid algorithms. Although ACO-BP algorithm is better than BP algorithm, it has slow convergence speed and is easy to fall into local optimization. In this study, when estimating $ET_O$ using ACO-BP model, the accuracy of individual stations is very low, and it is likely that the algorithm falls into local optimization. Moreover, ACO algorithm is prone to stagnation, that is, after the search is carried out to a certain extent, the solutions found by all individuals are completely consistent, and cannot further search the solution space, which is not conducive to finding better solutions. The convergence speed of CS algorithm has little correlation with parameter changes, and it is not easy to fall into local optimization. In this study, the estimation accuracy of CS-BP model is better than ACO-BP, and the solution of the model is very close to the standard value. Nazri Mohd. Nawi first proposed CS algorithm to optimize BP [27], and found that CS algorithm greatly improves the training efficiency of BP, which makes the result more accurate, which is consistent with the conclusion of this study. Among all hybrid algorithms, CSO-BP has the best performance and high goodness of fit. In recent years, some scholars have used CSO and PSO algorithms to optimize the prediction model [25], and found that the model optimized by CSO has more advantages. This shows that the optimization effect of CSO algorithm is very good, which is similar to the conclusion of this study. After inputting the main factors selected

by XGBoost algorithm, the four $ET_O$ models constructed in this study perform well, among which the evaluation indexes of CSO-BP (RMSE, $R^2$, MAE, NSE) ranging 0.200–0.0377, 0.932–0.984, 0.140–0.261, 0.920–0.984. The GPI range is 1.472–2.000 and the GPI ranking range is 1–23.

This shows that the factor combination selected by XGBoost algorithm is reliable. Input $T_{max}$, n and Ra into $ET_O$ model, and the accuracy loss is very small. Therefore, these three factors can be used as input reference for $ET_O$ model in Southwest China. In future research, we will combine satellite data to conduct a global-scale $ET_O$ model applicability study on the hybrid model.

## Conclusions

In this study, XGBoost algorithm is used to determine the combination of key factors affecting $ET_O$ prediction and determine as few key meteorological factors as possible. The combination of selected factors is used as inputs to construct $ET_O$ model based on machine learning and optimization algorithm (BP, ACO-BP, CSO-BP, CS-BP) in 12 stations of southwest China. The results showed that:

1. When the importance of meteorological factors to $ET_O$ model is determined by XGBoost algorithm, Tmax is the primary factor affecting $ET_O$. The sum of the importance of the top three factors ($T_{max}$, n and Ra) is greater than 82.9%, and these three factors can be used as the input combination of $ET_O$ model construction to save the calculation cost.

2. When BP algorithm is used to build $ET_O$ model, the prediction accuracy is satisfactory, with RMSE, $R^2$, MAE, NSE ranging 0.488–0.779, 0.499–0.930, 0.386–0.759, 0.400–0.905. The range of GPI value is -1.345–1.108, ranking 26–47. The model has the highest accuracy in Bazhong station, and GPI is 1.108, ranking 26.

3. When three optimization algorithms and BP are used to construct the hybrid model for predicting $ET_O$ (ACO-BP, CSO-BP, CS-BP), the hybrid model shows better accuracy than BP. Among them, CSO-BP model has the highest accuracy, with RMSE, $R^2$, MAE, NSE ranging 0.200–0.377, 0.932–0.984, 0.140–0.261, 0.920–0.984. The range of GPI value is 1.472–2.000, ranking 1–23. The pseudocode of the proposed model is provided in S3 Appendix.

## Supporting information

**S1 Appendix. Raw results of factor importance.**
(PDF)

**S2 Appendix. Raw results of the ETO model.**
(PDF)

**S3 Appendix. The pseudocode of the proposed model.**
(PDF)

## Author Contributions

**Conceptualization:** Zhenhua Zhao.

**Data curation:** Jing Zhang.

**Methodology:** Zhenhua Zhao.

**Software:** Jing Zhang.

**Supervision:** Zhenhua Zhao.

**Validation:** Zhenhua Zhao.

**Visualization:** Guohua Feng.

**Writing – original draft:** Guohua Feng.

**Writing – review & editing:** Zhenhua Zhao, Guohua Feng.

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
