## [Decision Letter · Decision Letter 0]

17 Jan 2022

PONE-D-21-39303The simplified hybrid model based on BP to predict the reference crop evapotranspiration in Southwest ChinaPLOS ONE

Dear Dr. Zhao,

Thank you for submitting your manuscript to PLOS ONE. After careful consideration, we feel that it has merit but does not fully meet PLOS ONE’s publication criteria as it currently stands. Therefore, we invite you to submit a revised version of the manuscript that addresses the points raised during the review process.

We look forward to receiving your revised manuscript.

Kind regards,

Vassilis G. Aschonitis

Academic Editor

PLOS ONE

Journal Requirements:

2. PLOS requires an ORCID iD for the corresponding author in Editorial Manager on papers submitted after December 6th, 2016. Please ensure that you have an ORCID iD and that it is validated in Editorial Manager. To do this, go to ‘Update my Information’ (in the upper left-hand corner of the main menu), and click on the Fetch/Validate link next to the ORCID field. This will take you to the ORCID site and allow you to create a new iD or authenticate a pre-existing iD in Editorial Manager. Please see the following video for instructions on linking an ORCID iD to your Editorial Manager account: https://www.youtube.com/watch?v=_xcclfuvtxQ.

3. We note that Figure 1 in your submission contain map image which may be copyrighted. All PLOS content is published under the Creative Commons Attribution License (CC BY 4.0), which means that the manuscript, images, and Supporting Information files will be freely available online, and any third party is permitted to access, download, copy, distribute, and use these materials in any way, even commercially, with proper attribution. For these reasons, we cannot publish previously copyrighted maps or satellite images created using proprietary data, such as Google software (Google Maps, Street View, and Earth). For more information, see our copyright guidelines: http://journals.plos.org/plosone/s/licenses-and-copyright.

a) You may seek permission from the original copyright holder of Figure 1 to publish the content specifically under the CC BY 4.0 license.  

4. Please ensure that you refer to Figures 2 and 3 in your text as, if accepted, production will need this reference to link the reader to the figures.

Reviewers' comments:

Reviewer's Responses to Questions

**Comments to the Author**

1. Is the manuscript technically sound, and do the data support the conclusions?

Reviewer #1: Yes

Reviewer #2: Yes

2. Has the statistical analysis been performed appropriately and rigorously? 

Reviewer #1: Yes

Reviewer #2: Yes

3. Have the authors made all data underlying the findings in their manuscript fully available?

Reviewer #1: Yes

Reviewer #2: Yes

4. Is the manuscript presented in an intelligible fashion and written in standard English?

Reviewer #1: Yes

Reviewer #2: No

5. Review Comments to the Author

Reviewer #1: The manuscript is well-written and the language is the appropriate one. Lines 38-40 is better ot be rewritten in order to be more comprhensive. The references need to be rewritten in the journal;s style. In some references are missing the year of the publication. It is a good research.

Reviewer #2: In this study, hybrid models for the computing of reference evapotranspiration in Southwest China are developed. The investigation is very useful and quite comprehensive as three optimization algorithms are used and the three hybrid models are compared with each other and with the machine learning algorithm. Also, the implementation is done on 12 meteorological stations which is positive. However, I have some concerns about the manuscript. The manuscript has problems in the use of English and several parts are difficult to understand. There are some format issues like some spaces are missing before brackets in the main text and in the References as well.

Here are only few examples:

38 – 40: “However, due to the uneven distribution and inconsistent scale of meteorological stations, which makes it difficult to accurately calculate ETO.” You could rephrase that.

59 – 62: I think some articles are missing. You could rephrase that.

63 – 64: “A more efficient prediction model with better adaptability is obtained.” It is not clear what you are referring to.

67: “Comprehensive meteorological parameters are difficult to obtain”. You should rephrase that.

69 – 71: “determine the dominant factors of ETO model based on path analysis theory, or use principal component analysis and factor analysis to select key determinants.” You could rephrase that.

103: I see 12 stations in the analysis.

118: " is the solar radiation of surface net", net solar radiation

Please correct the format of the units’ titles 2.3.1, 2.3.2 and 2.3.2.

Bold fonts are detected in cases where they shouldn't be (numbering of eq. (17 -21)).

There is inconsistency in the symbolism of solar radiation in eq. (1) and the rest of the manuscript.

299: “from the figure that the accuracy”. Table instead of figure

318 – 319: “In contrast, CSO-BP has the best performance in constructing ETO prediction model”. This sentence is not clear.

6. PLOS authors have the option to publish the peer review history of their article (what does this mean?). If published, this will include your full peer review and any attached files.

Reviewer #1: No

Reviewer #2: No

---

## [Author Response · Author response to Decision Letter 0]

1 Mar 2022

Dear editor,

Thank you for your consideration of this manuscript. We appreciate all valuable comments, and all revises are noted in the file named as “Revised Manuscript with Track Changes”. The reply to the request are as follow：

1. We rechecked and corrected the format of the manuscript.

2. The corresponding author verified and updated the ORCID ID （0000-0003-2018-9595）。

3. We have deleted Figure 1 from the manuscript.

4. We have added citations to Figs 2 and 3 in the paper.

Sincerely,

Zhenhua Zhao, Guohua Feng, Jing Zhang

Response to Comments

 Reviewer #1

Question1. Lines 38-40 is better to be rewritten in order to be more comprehensive.

Response：Thank you for your advice. As you suggested, the sentence has been revised. “However, due to the uneven distribution and inconsistent scale of meteorological stations, it is difficult to obtain all meteorological parameters used to calculate ETO, which makes it difficult to accurately calculate ETO.”. (Line45-Line46 in “Revised Manuscript with Track Changes”)

Question2. The references need to be rewritten in the journal’s style. In some references are missing the year of the publication.

Response：We revised the reference format according to the requirements of the journal and added the missing publication year.

Reviewer #2

Question1. The manuscript has problems in the use of English and several parts are difficult to understand. There are some format issues like some spaces are missing before brackets in the main text and in the References as well.

Response：Thank you for your valuable suggestions. We carefully reexamined our manuscript, found some language errors, and revised them. The spaces before the brackets have been added in the main text and in the references.

Question2.38–40: “However, due to the uneven distribution and inconsistent scale of meteorological stations, which makes it difficult to accurately calculate ETO.” You could rephrase that.

Response：The sentence has been revised to “However, due to the uneven distribution and inconsistent scale of meteorological stations, it is difficult to obtain all meteorological parameters used to calculate ETO, which makes it difficult to accurately calculate ETO.”. (Line45-Line46 in “Revised Manuscript with Track Changes”)

Question3. 59 – 62: I think some articles are missing. You could rephrase that.

Response：We added the missing sentence "and many scholars use optimization algorithms to improve the accuracy of models in recent years.", which made the content of the article more coherent. (Line69-Line71)

Question4.63 – 64: “A more efficient prediction model with better adaptability is obtained.” It is not clear what you are referring to.

Response：We have modified sentences with ambiguous semantics. “Fang et al. [18] used the fruit fly optimization algorithm (FOA) to optimize the ETO constructed by generalized regression neural networks (GRNN) algorithm, and obtained a more efficient and adaptive prediction model.” (Line71-Line74)

Question5. 67:“Comprehensive meteorological parameters are difficult to obtain”. You should rephrase that.

Response：The description in the manuscript has been changed to " There are many difficulties in the actual collection of meteorological data, which makes it difficult to obtain all meteorological parameters used to calculate ETO,". (Line78-Line80)

Question6. 69 – 71: “determine the dominant factors of ETO model based on path analysis theory, or use principal component analysis and factor analysis to select key determinants.” You could rephrase that.

Response：The description in the manuscript has been changed to " determine the dominant factors of ETO model based on path analysis theory, principal component analysis and factor analysis." (Line83-Line84)

Question7. 103: I see 12 stations in the analysis.

Response：The number of sites has been modified. (Line116)

Question8. 118: " is the solar radiation of surface net", net solar radiation

Response：The description has been modified to "net solar radiation". (Line132)

Question9. Please correct the format of the units’ titles 2.3.1, 2.3.2 and 2.3.2.

Response：The format of the title has been modified according to the requirements of the journal.

Question10. Bold fonts are detected in cases where they shouldn't be (numbering of eq. (17 -21)).

Response：The wrong font has been modified.

Question11. There is inconsistency in the symbolism of solar radiation in eq. (1) and the rest of the manuscript.

Response：The symbolism of solar radiation in eq. (1) is net solar radiation (Rn) and the rest of the manuscript is extraterrestrial solar radiation (Ra). We have concretized the definition of Ra in the paper. (Line 125)

Question12. 299: “from the figure that the accuracy”. Table instead of figure.

Response：The word "figure" has been replaced with "table". (Line 323)

Question13. 318 – 319: “In contrast, CSO-BP has the best performance in constructing ETO prediction model”. This sentence is not clear.

Response：The description in the manuscript has been changed to “Among all ETO models, CSO-BP model has the best accuracy advantage in estimating ETO at each site.” (Line342-Line344) 

We tried our best to improve the manuscript. Changes were marked in red in revised paper (Revised Manuscript with Track Changes). We appreciate for Editors/Reviewers’ warm work earnestly, and hope that the correction will meet with positive responses. Once again, many special thanks for your good comments and suggestions!

---

## [Decision Letter · Decision Letter 1]

4 May 2022

PONE-D-21-39303R1The simplified hybrid model based on BP to predict the reference crop evapotranspiration in Southwest ChinaPLOS ONE

Dear Dr. Zhao,

Thank you for submitting your manuscript to PLOS ONE. After careful consideration, we feel that it has merit but does not fully meet PLOS ONE’s publication criteria as it currently stands. Therefore, we invite you to submit a revised version of the manuscript that addresses the points raised during the review process.

We look forward to receiving your revised manuscript.

Kind regards,

Ardashir Mohammadzadeh, Phd

Academic Editor

PLOS ONE

Journal Requirements:

Additional Editor Comments (if provided):

Add the condition of compared methods; Add the pseudo code of the suggested approach; To give a better direction for readers add some remarks about potential improvement by the use of type-3 fuzzy logic systems and new learning methods;

Reviewers' comments:

Reviewer's Responses to Questions

**Comments to the Author**

1. If the authors have adequately addressed your comments raised in a previous round of review and you feel that this manuscript is now acceptable for publication, you may indicate that here to bypass the “Comments to the Author” section, enter your conflict of interest statement in the “Confidential to Editor” section, and submit your "Accept" recommendation.

Reviewer #1: All comments have been addressed

2. Is the manuscript technically sound, and do the data support the conclusions?

Reviewer #1: Yes

3. Has the statistical analysis been performed appropriately and rigorously? 

Reviewer #1: Yes

4. Have the authors made all data underlying the findings in their manuscript fully available?

Reviewer #1: Yes

5. Is the manuscript presented in an intelligible fashion and written in standard English?

Reviewer #1: Yes

6. Review Comments to the Author

Reviewer #1: No further comments. They are all adressed in the manuscript. The article can be accepted for publication.

7. PLOS authors have the option to publish the peer review history of their article (what does this mean?). If published, this will include your full peer review and any attached files.

Reviewer #1: No

---

## [Author Response · Author response to Decision Letter 1]

11 May 2022

Dear editor,

Thank you for your consideration of this manuscript. We appreciate all valuable comments. We have uploaded our figure files to the Preflight Analysis and Conversion Engine (PACE) digital diagnostic tool, and resubmitted the figures in the correct format. Each comment is responded below, and all revises are noted in the file named as “Revised Manuscript with Track Changes”.

Response to Comments

Question1. Add the condition of compared methods.

Response: We thank the editor for the constructive comments. We supplemented the parameter conditions of compared methods in the manuscript: (1) BP algorithmml: training times (net.trainParam.epochs=1000), learning rate (net.trainParam.lr=0.01), Minimum error of training goal (net.trainParam.goal=0.00001), momentum factor (net.trainParam.mc=0.01); (2) ACO algorithmml: population size (popsize=10), maximum Generation (maxgen=50), pheromone polatility (rou=0.9), transition probabilities constant (p0=0.2); (3) CSO algorithmml: population size(popsize=10), maximum iterations(npop=50), mixture ratio (MR=0.3); (4) CS algorithmml: initial population size (PopulationSize_Data=30), probability of cuckoo eggs being found (pa = 0.25), step control amount (cs_alpha=1.0).” (Line143-146, Line 192-194, Line 214-215, Line 254-256)

Question2. Add the pseudo code of the suggested approach.

Response: We supplement the pseudo code of the recommendation model (CSO-BP) in the supporting file. The pseudocode of the proposed model is provided in S3 Appendix. (Line422-423)

“Input: 

Population size sizepop

Maximum number of iterations maxiter

The number of input parameter inputnum

The number of output parameter outputnum

BP structure net

Number of samples sample

Range of genetic changes popmax

Fitness function fun

Upper and lower limits of speed Vmax, Vmin

Function CSO_ITR(maxiter, sizepop):

 Initialize:

for i in 1:sizepop do

 pop[i,:] ⬅popmax*rands(1, sample) //generate random number in [-5,5]

 V[i,:] ⬅rands(1,sample)

fitness[i]⬅fun(pop[i,:],inputnum,hiddennum,outputnum,net)

end

 fitnessgbest ⬅record the current individual extreme value

fitnesszbest⬅record the current population extreme value

Iterative Refinement:

 for i in 1:maxiter do

 for j in 1:sizepop do

 //update velocity

 V[j,:]⬅V[j,:] + c1*rand*(gbest[j,:]- pop[j,:]) + c2*rand*(zbest – pop[j,:])

 Limit V to [Vmin, Vmax]

 // update population

 pop[j,:] ⬅pop[j,:]+0.2*V[j,:]

 Limit pop to [popmin, popmax]

 Adaptive variation

 Update Fitness value

 end

 for k in 1:sizepop do

 //Individually optimal update

 if fitness[k] < fitnessgbest[k] then

 gbest[k,:] ⬅pop[k,:]

 fitnessgbest[k] ⬅fitness[k]

 end

 // group optimal update

if fitness[k]<fitnesszbest[k] then

zbest⬅pop[k,:]

fitnesszbest=fitness[k]

end

 end

// The optimal value zbest of the swarm algorithm is used as the initial weight of the network BP

Train(net)

return net(test_data)”

Question3. To give a better direction for readers add some remarks about potential improvement by the use of type-3 fuzzy logic systems and new learning methods.

Response: In further research, the study scale could serve as a potential improvement. We have supplemented the description in the manuscript. “In future research, we will combine satellite data to conduct a global-scale ETO model applicability study on the hybrid model.” (Line 402-404)

”

---

## [Decision Letter · Decision Letter 2]

27 May 2022

The simplified hybrid model based on BP to predict the reference crop evapotranspiration in Southwest China

PONE-D-21-39303R2

Dear Dr. Zhao,

We’re pleased to inform you that your manuscript has been judged scientifically suitable for publication and will be formally accepted for publication once it meets all outstanding technical requirements.

Kind regards,

Ardashir Mohammadzadeh, Phd

Academic Editor

PLOS ONE

Additional Editor Comments (optional):

Reviewers' comments:

Reviewer's Responses to Questions

**Comments to the Author**

1. If the authors have adequately addressed your comments raised in a previous round of review and you feel that this manuscript is now acceptable for publication, you may indicate that here to bypass the “Comments to the Author” section, enter your conflict of interest statement in the “Confidential to Editor” section, and submit your "Accept" recommendation.

Reviewer #1: All comments have been addressed

2. Is the manuscript technically sound, and do the data support the conclusions?

Reviewer #1: Yes

3. Has the statistical analysis been performed appropriately and rigorously? 

Reviewer #1: Yes

4. Have the authors made all data underlying the findings in their manuscript fully available?

Reviewer #1: Yes

5. Is the manuscript presented in an intelligible fashion and written in standard English?

Reviewer #1: Yes

6. Review Comments to the Author

Reviewer #1: (No Response)

7. PLOS authors have the option to publish the peer review history of their article (what does this mean?). If published, this will include your full peer review and any attached files.

Reviewer #1: No

---

## [Editor Report · Acceptance letter]

1 Jun 2022

PONE-D-21-39303R2 

The simplified hybrid model based on BP to predict the reference crop evapotranspiration in Southwest China 

Dear Dr. Zhao:

I'm pleased to inform you that your manuscript has been deemed suitable for publication in PLOS ONE. Congratulations! Your manuscript is now with our production department. 

Kind regards, 

on behalf of

Dr. Ardashir Mohammadzadeh 

Academic Editor

PLOS ONE